# Prevention of Fungal Contamination in Semi-Hard Cheeses by Whey–Gelatin Film Incorporated with *Levilactobacillus brevis* SJC120

**DOI:** 10.3390/foods12071396

**Published:** 2023-03-25

**Authors:** Sofia P. M. Silva, José A. Teixeira, Célia C. G. Silva

**Affiliations:** 1Institute of Agricultural and Environmental Research and Technology (IITAA), University of the Azores, 9700-042 Angra do Heroísmo, Portugal; 2Centre of Biological Engineering (CEB), University of Minho, Campus Gualtar, 4710-057 Braga, Portugal; 3LABBELS-Associate Laboratory, University of Minho, Campus Gualtar, 4710-057 Braga, Portugal

**Keywords:** cheese whey, lactic acid bacteria, antifungal activity, cheese, edible film

## Abstract

Cheese whey fermented by lactic acid bacteria (LAB) was used to develop an edible film with antifungal properties. Five LAB strains isolated from artisanal cheeses were screened for antifungal activity and incorporated into a whey–gelatin film. Of the strains tested, *Levilactobacillus brevis* SJC120 showed the strongest activity against five filamentous fungi isolated from cheese and cheese-making environment, at both 10 °C and 20 °C. The cell-free supernatant from *L. brevis* inhibited fungal growth by more than 80%. Incorporation of bacterial cells into the film did not alter the moisture content, water vapor permeability, or mechanical and optical properties. The whey–gelatin film was also able to maintain the viability of *L. brevis* cells at 10^7^ log CFU/g after 30 days at 10 °C. In cheeses wrapped with *L. brevis* film, the size of fungal colonies decreased by 55% to 76%. Furthermore, no significant differences (*p* > 0.05) were observed in cheese proteolysis or in the moisture, fat, and protein content of the cheese wrapped with films. The results showed that whey–gelatin film with *L. brevis* SJC120 can reduce the contamination of cheese with filamentous fungi and could be used as an alternative to conventional cheese preservation and packaging.

## 1. Introduction

Fungal spoilage causes visible or invisible sensory defects in cheese, such as fungal growth on the surface and the production of metabolites that cause unpleasant changes in aroma, taste, and texture, leading to a loss of product quality [1]. In the dairy industry, fungi responsible for problems in cheese production are diverse and belong to several genera such as *Penicillium, Aspergillus*, *Cladosporium*, *Geotrichum*, *Mucor*, and *Trichoderma* [2,3]. In addition to the major economic losses associated with spoilage, some fungi pose a food safety hazard due to their ability to produce mycotoxins. The risk of mycotoxins in cheese increases when toxigenic fungi such as *Aspergillus* and *Penicillium* are allowed to grow during cheese production and storage [1]. The growth of fungi in cheese is largely due to their ability to grow at refrigeration temperatures, low pH, low oxygen tensions, lipolytic activity, and resistance to weak acid preservatives [4]. Fungal contamination of cheese can occur at various stages of production, with the air in ripening rooms being one of the main sources of contamination [5,6,7]. 

To prevent food spoilage by fungi during cheese production, the use of antifungal additives, such as sorbates, benzoates, and natamycin, is the most common industry response to minimize food spoilage and the resulting economic losses [8,9]. However, consumers are demanding new alternatives for foods without additives or foods based on natural ingredients [10]. Some species/strains of lactic acid bacteria (LAB) can be used as bioprotective agents [11], as they have been shown to retard fungal growth [12] by producing a variety of active antagonistic metabolites that include organic acids [13,14,15], proteinaceous compounds [16,17,18], fatty acids [19], and bacteriocin-like substances [20].

Edible films and coatings have been shown to be excellent biomaterials for use as carriers for LAB strains [21,22,23,24], providing good protection against LAB loss of viability and maintaining adequate concentrations of antimicrobial compounds on the food surface. For example, the incorporation of live LAB into edible films and coatings with antimicrobial activity has shown positive results in inhibiting spoilage bacteria when applied to a food product [25,26,27,28,29]. The use of edible films and coatings has also attracted considerable interest due to their edibility, biodegradability, and use for various purposes in the food industry [30]. Many types of polymeric materials have been used in the development of edible films and coatings with various functional properties, especially as effective carriers for bioactive compounds [31,32,33]. Among the various formulations, collagen (gelatin) and whey proteins have been widely used for incorporating live bacteria such as LAB into edible packaging [32,34,35,36]. 

Several authors have investigated the use of pure cheese whey (CW) as a nutrient medium for the growth of LAB [37,38,39,40], showing the versatility of LAB strains in adapting to different culture conditions and that CW supports well the growth of LAB. Whey is the most important by-product of cheese production; the processing of this waste is a problem for the dairy industry, because it is produced in large quantities during cheese-making and has a high contaminant load [41,42]. However, it can be reused for biotechnological purposes, because CW contains functional proteins and peptides, lipids, lactose, minerals, and vitamins. This source of bioactive whey proteins and peptides can be used as a substrate for the growth of LAB and for the production of edible films. Some studies have reported that edible films containing LAB fermented whey have also shown beneficial effects on reducing pathogenic bacteria [34] and inhibiting contamination by fungi [35]. However, to our knowledge, there is limited work on the incorporation of LAB into edible films and coatings to control fungal contamination [36,43,44,45], and few have investigated these films in cheese products [35,46].

The aim of this study was to develop an active gelatin film containing LAB-fermented whey for use as an antifungal biopreservative for semi-hard cheeses. To achieve this goal, five LAB strains were incorporated into the whey–gelatin film and tested against five filamentous fungi for 15 days at 10 °C and 20 °C to select a strain with high antifungal activity. In further experiments, the cell-free supernatant (CFS) produced by the selected strain was tested against the radial fungal growth of the indicator fungus. In addition, the influence of the bacterial cells added to the films was investigated with respect to (a) the physicochemical and structural properties of the film, (b) the changes in cheese proteolysis and composition during ripening, and (c) the activity of the films against *Penicillium* spp. contamination of cheese.

## 2. Materials and Methods

### 2.1. Microorganisms and Growth Conditions

*Lactococcus lactis* subsp. *lactis* L3B1M7 (GenBank KM079358), *Latilactobacillus curvatus* SJC53 (GenBank MT742887), *Lacticaseibacillus paracasei* SJC87 (GenBank MT742921), *Lacticaseibacillus casei* SJC88 (GenBank MT742922), and *Levilactobacillus brevis* SJC120 (GenBank OQ457279) were previously isolated from artisanal cheeses from the Azores [47] and selected for antifungal activity [48]. LAB strains kept at −80 °C were revitalized and propagated in MRS broth (Biokar Diagnostics, France) at 30 °C before use.

*Aspergillus chevalieri* MUM 00.07, *Aspergillus flavus* MUM 16.106, *Penicillium brevicompactum* MUM 9906, *Penicillium commune* MUM 16.56, and *Penicillium nordicum* MUM 16.93 were isolated from cheese rind and cheese-ripening chamber and kindly supplied by the Micoteca da Universidade do Minho (MUM). These strains were used as target fungi and grown in potato dextrose agar (PDA, HiMedia Laboratories, India) in the dark at 25 °C. To determine the spore concentration in the inoculum, spores were counted under the microscope in a Neubauer cell chamber and adjusted to 10^4^ spores/mL.

Cheese whey was used as a substrate for LAB fermentation and for the incorporation of an edible film. Whey was prepared from whole milk (protein: 3.43% *w/w*, total fat: 4.26% *w/w*) from Chegalvorada (dairy farm of the University of the Azores, Angra do Heroísmo, Portugal) as described by Silva et al. [34]. The whey was pasteurized (65 °C for 30 min), stored at 4 °C, and used within 24 h. 

### 2.2. Production of Films

To produce fermented whey, a pre-inoculum was obtained by culturing each LAB in MRS broth at 30 °C overnight. The bacterial cultures (20 mL) were then centrifuged at 3600× *g* for 20 min at 20 °C (Centrifuge 5804R, Eppendorf, Hamburg, Germany), washed with PBS, and the pellet with cells was added to 100 mL of pasteurized cheese whey. The inoculated whey was incubated at 30 °C for 48 h. Two types of whey were prepared: LAB fermented whey (FW) and control whey without addition of LAB (CW). The pH of CW (without addition of LAB) was adjusted to 3.5 with 0.1 mol/mL HCL to achieve a similar pH in the fermented whey.

Gelatin (VWR, USA) was used as the support material for the film formulation, while glycerol (HiMedia Laboratories, India) was used as the plasticizer. All materials, except gelatin, were previously sterilized at 121 °C for 15 min. To prepare the films, 6 g of gelatin and 2 g of glycerol were dissolved in 100 mL of FW and CW. The final concentration of LAB cells in the film solution was 10^11^ CFU/L. The film solution was magnetically stirred and heated to 30 °C to improve the homogeneity of the films. The film solutions were poured onto Teflon casting plates and dried at 20 °C, as described by Silva et al. [34].

### 2.3. Antifungal Activity of Films with LAB-Fermented Whey

This preliminary experiment was designed to evaluate the antifungal activity of whey–gelatin films, in which different LAB strains were incorporated to select the most effective one.

The antifungal activity of the films was evaluated as described by Guimaraes et al. [49] with some modifications. Five films prepared with FW from each LAB (*Lc. lactis* L3B1M7, *L. curvatus* SJC53, *L. paracasei* SJC87, *L. casei* SJC88, and *L. brevis* SJC120) were tested to assess their efficacy against the target fungi. Films with CW (without any LAB) were used as controls. Film circles (35 mm diameter) were placed in Petri plates containing PDA media, and 10 µL of a spore solution (10^4^ spores/mL) of each target fungus was inoculated centrally onto the surface of the film. The plates were incubated at 10 °C and 20 °C for 15 days. The two opposite diameters of the growing fungal colonies were measured after 5, 10, 12, and 15 days. All experiments were performed in triplicate. 

### 2.4. Characterization of the Antifungal Properties of Cell-Free Supernatant

The antifungal properties of cell-free supernatant (CFS) of *L. brevis* SJC120 was tested as described by Guimarães et al. [49]. *L. brevis* SJC120 was cultured in pasteurized cheese whey at 30 °C for 48 h, as previously described. Cells were removed by centrifugation (3600× *g* for 20 min at 20 °C), and the resulting CFS was added to PDA medium (10% (*v/v*) and poured into Petri plates). After cooling, the plates were inoculated in the center with 10 µL of each fungal suspension (10^4^ spores/mL). In the control experiments, cheese whey was added instead of CFS. The plates were incubated for 7 days at 25 °C in the dark. The opposite diameters of the growing fungal colonies were measured at the end of the incubation period. 

To evaluate the effect of pH neutralization, CFS was adjusted to pH 7 with 1 M NaOH. To determine the nature of the potential antifungal compounds in CFS produced by *L. brevis* SJC120, 10 mL of neutralized CFS was treated with 2 mg/mL proteinase K (Sigma-Aldrich, Mannheim, German) and incubated at 37 °C for 3 h. The samples were boiled for 5 min to inactivate the enzyme [50]. All experiments were performed in triplicate. Radial growth of fungi was measured (mm) [51]. 

### 2.5. Enumeration of LAB in the Films

The viability of *L. brevis* SJC120 was evaluated in films stored at 10–13 °C and 65–70% relative humidity (RH), at regular intervals: 3, 5, 10, 15, 20, 25, and 30 days. Films were cut into circular shapes (20 mm diameter at approximately 0.1 g) and placed in a sterile bag containing 9.9 mL phosphate buffer solution (PBS) (0.01 M Na_2_HPO_4_/NaH_2_PO_4_, pH 7.4) and homogenized in a stomacher (3 min at 260 rpm, 400 Circulator Lab Blender, Seward, UK), according to Guimarães et al. [46]. Sequential tenfold dilutions were made with peptone water (Biokar Diagnostics, France) and plated on MRS agar (Biokar Diagnostics, France). Plates were incubated at 30 °C for 48–72 h. LAB colonies were counted, and results were expressed as Log CFU/g. All experiments were performed in triplicate. 

### 2.6. Film Characterization

Characterization of the film in terms of thickness, moisture content and water solubility, water vapor permeability, limonene permeability, tensile strength and elongation at break, and optical properties with *L**, *a**, and *b** parameters and opacity were performed as described by Silva et al. [34].

### 2.7. Antifungal Activity of Films for the Control of Fungal Growth in Cheese

Cylindrical and semi-hard bovine cheddar cheese was prepared at the University of the Azores from whole milk (gross composition: 3.4% protein, 4.3% fat) from the university farm (Angra do Heroísmo, Portugal), as described by Silva et al. [34]. Cheeses were stored for one day at 10–13 °C and 65–70% RH before the film was applied.

For the experiments, cheese samples weighing approximately 16 g and measuring 6 cm in diameter and 0.5 cm in thickness were covered with squares of each film (10 × 10 cm), ensuring that the entire surface was covered. Target fungi *P. brevicompactum*, *P. commune*, and *P. nordicum* (10^4^ spores/mL) were inoculated onto the cheese covered with the film to determine their antifungal activity. This situation mimicked the hypothetical contamination of the cheese by fungi.

All cheese manipulations and film application were performed under sterile conditions. After inoculation, cheeses were ripened at 10–13 °C and 65–70% RH. Fungal growth was evaluated by measuring the two opposite diameters of the fungal colonies after 5, 10, 15, 20, 25, and 30 days. The films with *L. brevis* SJC120 and the films without LAB cells (control) were tested in triplicate for each fungus.

### 2.8. Cheese Analysis 

Twenty-seven experimental cheeses were made on three different occasions, as described previously (Section 2.7). For each cheese, approximately 400 g of curd was molded into cheese molds (100 mm in diameter and 40 mm high). Squares (20 × 20 cm) of each film were used to cover the entire surface of the cheese. Within each batch, cheeses were randomly assigned to three treatments: (1) uncovered cheese (UN); (2) covered with whey–gelatin film (without LAB) (CW); (3) covered with whey–gelatin film with incorporated *L. brevis* SJC120 (W120). The cheeses were ripened for 30 days at 10–13 °C and 65–70% RH.

The gross composition of the cheese was determined in grated samples after removal of the films. Cheeses were analyzed in triplicate for their moisture content [52], fat [53], and protein [54]. Analyses were carried out at the end of ripening (30 days).

Proteolysis was assessed by analyzing the nitrogen and free amino acid content in the water-soluble extract (WSE) of the cheeses. To prepare the WSE, 20 g of grated cheese was homogenized in 100 mL of Milli-Q water using a stomacher (5 min at 260 rpm, 400 Circulator Lab Blender, Seward, UK), followed by heating at 40 °C for 60 min and centrifugation at 4 °C for 30 min at 2000× *g* (Beckman Coulter J2-HC, Krefeld, Germany). The fat layer was removed and the supernatant was filtered through Whatman No.1 filter paper. Total nitrogen content (TN) was determined by the Kjeldahl method using a Kjeltec System 2300 distillation apparatus (Tecator Technology, Foss, Spain). Total free amino acids (FAA) were determined on WSE using the cadmium-ninhydrin method described by Radeljević et al. [55]. The WSE (50 µL) was diluted to 1 mL with distilled water and mixed with 2 mL of Cd-ninhydrin reagent containing 0.8 g of ninhydrin (Merck, Germany), 80 mL of 99.5% ethanol (VWR, La Chapelle-sur-Erdre, France), 10 mL of acetic acid (Chem-Lab, Zedelgem, Belgium), and 1 g of cadmium chloride (Sigma-Aldrich, Saint Louis, MO, USA). After heating at 84 °C for 10 min and cooling to room temperature, the absorbance was measured at 507 nm (Genesys 30 spectrophotometer, Thermo Fisher Scientific Inc., Madison, WI, USA). The concentration of free amino groups in the mixture was expressed as m*M* leucine (m*M* Leu) using a standard curve of leucine (Fluka, Darmstadt, Germany).

Texture profile analysis was performed according to the method described by Primo-Martín et al. [56] with slight modifications. Cheese samples were cut into 40 mm high wedges and stored at room temperature for 1 h before testing. The firmness of the cheese samples was determined after 0, 15, and 30 days of ripening using a texturometer (TMS-PRO, Food Technology Corporation, Sterling, VA, USA). The cheese samples were fractured using a wedge-shaped aluminium probe (30° cutting angle, 25 mm wide) on a table with a slit (width 90 mm, length 100 mm) at a speed of 100 mm/min.

All experiments for texture, WSE, and cd-ninhydrin were performed in triplicate at 0, 15, and 30 days after cheese manufacture.

The color determination of the cheeses covered with the films was carried out as previously described (Section 2.6) at 0, 15, and 30 days after cheese making. To evaluate the effect of the film on the color of the cheese, the films were removed on the 15th and 30th day of ripening and the opacity of the films was measured in triplicate during the same period.

### 2.9. Statistical Analysis

Results are expressed as means ± standard error of the mean (SEM). Mean results of antifungal activity were compared by one-way analysis of variance (ANOVA). Over time data were compared by ANOVA factorial. Post hoc Tukey’s test was used to discriminate between treatments whenever ANOVA detected significant differences (*p* < 0.05). All statistical analysis were performed with IBM SPSS Statistics, version 28 (IBM Corporation, New York, NY, USA).

## 3. Results and Discussion

### 3.1. Antifungal Activity of Films with LAB-Fermented Whey

The antifungal potential of the five LAB strains (*L. lactis* L3B1M7, *L. curvatus* SJC53, *L. paracasei* SJC87, *L. casei* SJC88, and *L. brevis* SJC120) was tested by incorporation into whey–gelatin films at 10 °C and 20 °C. Figure 1 and Figure 2 show the antifungal activity of whey–gelatin films containing the LAB strains at 10 °C and 20 °C, respectively, after inoculation with filamentous fungi. As the films were developed to protect cheese, the species tested were selected from those isolated from the cheese rind and the cheese-making environment. However, no germination of conidiospores was observed at 10 °C for the *Aspergillus* strains (*A. chevalieri* MUM 00.07 and *A. flavus* MUM 16.106), as *Aspergillus* spp. Have difficulty growing at refrigeration temperatures [6]. Therefore, only the germination of conidiospores of *Penicillium* species (*P. brevicompactum* MUM 9906, *P. commune* MUM 16.56, and *P. nordicum* MUM 16.93) is shown in Figure 1.

At 10 °C, whey–gelatin films containing *L. brevis* SJC120 showed a significant reduction (*p* < 0.05) in the growth of *P. brevicompactum* compared to the control after 5–15 days of storage (Figure 1A). In addition, *L. brevis* SJC120 completely prevented the growth of *P. commune* (Figure 1B). For *P. nordicum* (Figure 1C), *L. brevis* SJC120 also proved to be the LAB strain with the highest inhibition of fungal growth (*p* < 0.05). Other LAB strains showed slight inhibition of filamentous fungi growth, but not significantly different from control (*p* > 0.005).

At 20 °C, all target fungi grew well in the control films (without LAB cells) during the 15 days. At this temperature, more LAB species showed a significant reduction in the diameter of colonies of *Penicillium* species than at 10 °C. These results are to be expected, as LAB can grow better at 20 °C, resulting in more competitive growth and producing a greater amount of antifungal compounds. Each LAB also showed different behavior depending on the fungus. 

At the end of the storage period, the reduction of *A. chevalieri* (Figure 2A) was higher in whey–gelatin films prepared with *L. curvatus* SJC 53 (6.267 ± 0.567 mm), *L. paracasei* SJC 87 (5.867 ± 0.475 mm), and *L. casei* SJC 88 (15.750 ± 1.600 mm) compared to control films (41.600 ± 1.796 mm) and to films containing *Lc. lactis* L3B1M7 (52.217 ± 5.715 mm). Moreover, the whey–gelatin film containing *L. brevis* SJC120 completely inhibited the growth of this fungus. Like *A. chevalieri*, three strains (*L. curvatus* SJC 53, *L. paracasei* SJC 87, and *L. casei* SJC 88) also showed strong antifungal activity (*p* < 0.05) against *A. flavus* (Figure 2B). In addition, films with *L. brevis* SJC120 suppressed the growth of this fungus for up to 15 days. For *P. brevicompactum*, all five tested LAB showed an inhibitory effect compared to the control (*p* < 0.05) (Figure 2C), although none could completely suppress fungal growth. In contrast, the *L. brevis* SJC120 films completely inhibited the growth of *P. commune* (Figure 2D), unlike the other LAB tested, which showed no significant reduction (*p* > 0.05) of this fungus compared to the control. In addition, all LAB, which were incorporated in whey–gelatin film, strongly inhibited the growth of *P. nordicum* (*p* < 0.05) compared to the control, from the 10th day of storage at 20 °C (Figure 2E). On day 15, *L. brevis* SCJ 120 showed strong inhibition of fungal growth (9.6 ± 1.2 mm), while other strains showed medium inhibition (17.833 ± 1.433 mm), from 13.667 ± 1.636 mm of *L. curvatus* SJC 53 to 23.433 ± 3.844 mm of *L. paracasei* SJC 87, compared to control films (52.817 ± 3.962 mm) (Figure 2E).

Overall, the results showed that storage temperature affects antifungal activity, especially that of *L. lactis* L3B1M3. For *Penicillium* spp., no antifungal activity was observed at 10 °C, and inhibition of the growth of the same fungi was detected at 20 °C. *L. curvatus* SJC 53, *L. paracasei* SJC 87, and *L. casei* SJC 88 showed intermediate and similar antifungal activity, and no effect of temperature was observed. The antifungal activity of *L. brevis* SJC120 was strong and prevented the growth of *A. chevalieri*, *A. flavus*, and *P. commune* at 10 °C and 20 °C and reduced the radial growth of *P. brevicompactum* and *P. nordicum* at both temperatures. The described antifungal activity of *L. brevis* SJC120 is similar to previous studies confirming the antifungal capacity of *L. brevis* spp. [57,58,59]. Therefore, of the five LAB bacteria, *L. brevis* SJC120 was selected for subsequent application in whey–gelatin films.

### 3.2. Characterization of the Antifungal Properties of CFS

The antifungal activity of LAB can be achieved either by direct competition between cells or by the active metabolites produced by LAB. Therefore, we tested the inhibitory effect of the metabolites produced by *L. brevis* SJC120 (CFS) on the growth of the five fungi: *A. chevalieri* MUM 00.07, *A. flavus* MUM 16.106, *P. brevicompactum* MUM 9906, *P. commune* MUM 16.56, and *P. nordicum* MUM 16.93 (Figure 3).

Without any treatment, the antifungal activity of CFS of *L. brevis* SJC120 showed a strong inhibitory effect on the growth of five fungi (*p* < 0.05). It completely inhibited the growth of *A. chevalieri* (Figure 3A), *P. brevicompactum* (Figure 3C) and *P. commune* (Figure 3D) and reduced the growth of *A. flavus* and *P. nordicum* by 83% and 71%, respectively (Figure 3B,E). A strong antifungal activity of *L. brevis* strains (inhibition rate of about 80% by CFS) was also reported in other studies [60].

To rule out the effect of organic acids in inhibiting fungal growth, the neutralized CFS was further tested. The results showed similar inhibition of neutralized CFS (CFS pH 7) against four indicator fungi (*A. chevalieri*, *A. flavus*, *P. brevicompactum*, and *P. commune*) as with untreated CFS. However, the antifungal activity of CFS was reduced at neutral pH 7, resulting in a decrease in inhibitory activity from 71% to 46% for *P. nordicum*. This result suggests that the acids produced by *L. brevis* SJC120 are partly responsible for the inhibitory activity against *P. nordicum*. Other studies also reported that CFS was active against filamentous fungi only in a low pH environment [51,60,61]. Lactic acid, acetic acid, and phenyllactic acid were identified as responsible for the antifungal activity of CFS from *L. brevis* strains at low pH [61]. The antifungal activity of the organic acids produced by these strains occurs after they have penetrated cell membranes in their undissociated form, resulting in a decrease in intracellular pH and disruption of metabolic activities [18].

In the present study, treatment with the protease K of CFS from *L. brevis* SJC120 showed a reduction in inhibitory activity, especially against *P. brevicompactum, P. commune*, and *P. nordicum* (Figure 3C–E). These results suggest that the antifungal activity of this strain is not only due to the production of organic acids, but also to the production of proteinaceous compounds. The strong antifungal activity of *L. brevis* SJC120 could be due to the combined action of the bioactive compounds (organic acids and peptides) contained in CFS. Gerez et al. [61] also observed a decrease in the antifungal activity of *L. fermentum* after treatment with trypsin, proteinase K, and pepsin. Other authors have reported the production of proteinaceous antifungal substances from *L. brevis* [59,62]. 

### 3.3. L. brevis SJC120 Viability in the Film

The viability of *L. brevis* SJC120 on whey–gelatin films was investigated during the storage period of 30 days at 10 °C (Figure 4) to evaluate the ability of the film to ensure the viability and stability of the bacterial cells.

Starting from an initial bacterial cell concentration of about 10^11^ Log CFU/g, *L. brevis* SJC120 counts remained constant for 15 days. Between the 15th and 20th day, a considerable decrease in viability of 2 Log CFU/g was observed. From day 20 to day 30, the decrease in viability is less pronounced (about 1 Log CFU/g). The decrease in viable cells of *L. brevis* SJC120 by about 4 Log CFU/g throughout the storage period may be due to nutrient depletion and desiccation of the film. Similar studies on the survival of bacteria in films were conducted by Sánchez et al. [63]. They reported a decrease in the viability of *L. plantarum* in different biopolymer matrices during the storage period at 5 °C but were able to maintain at least half of the initial number of viable cells. Pereira et al. [22] also showed that the number of viable cells of *B. animalis* and *L. casei* in whey-protein-based films decreased after 60 days of storage at 4 °C, although the viability of both probiotic species decreased notably in the film systems stored at 23 °C.

The viability of bacterial cells in an edible film depends on many factors, such as the type of immobilized strain, the lack of nutrients during the storage process, the desiccation of the film, and the presence of oxygen, but the temperature during storage and the composition of the films also play an important role in the survival and stability of the cells [24,64,65]. At lower temperatures, a higher viability of the LAB cells is achieved, because at lower temperatures, the reaction rate decreases and the microorganisms are kept in a latent state, avoiding rearrangements in the wall material and thus preventing insufficient exposure of the microorganism [23]. This is in agreement with Soukoulis et al. [24], who also reported that the viability of *L. rhamnosus* immobilized in edible films for 25 days of storage decreased at 4 °C and 25 °C, with this decrease being more pronounced at higher temperatures. Similarly, Silva et al. [29] observed a constant viability of *L. lactis* and *L. garvieae* incorporated in an alginate–maltodextrin–glycerol matrix at 4 °C and 10 °C for 10 days.

The use of whey proteins in the composition of films has a positive effect on bacterial viability by providing nutrients to the cells, reducing the redox potential of the medium and increasing the buffering capacity of the medium, resulting in a lower drop in pH [22,66]. Guimarães et al. [46] reported the ability of whey protein films to maintain half the population of *L. buchneri* after 30 days of storage, even at 25 °C. The use of whey proteins was reported to have a positive effect on the viability of LAB [67]. Whey proteins showed no acute toxic effects on the viability of *L. rhamnosus* cells, neither during film production nor during the storage period [66]. Furthermore, gelatin was associated with the highest protective effect against osmosis during the film drying process. Proteins have been shown to enhance the survival of LAB by scavenging free radicals and providing peptides and amino acids essential for bacterial growth [65].

Thus, the use of cheese whey as a substrate for the growth of *L. brevis* SJC120 and as a basis for the formulation of a gelatin film appears to be a favorable environment for the sustainability of this LAB strain during cheese ripening at lower temperatures.

### 3.4. Film Characterization

The physicochemical properties (thickness, moisture content, solubility, water vapor transmission rate (WVTR), water vapor permeability (WVP), limonene permeability (LP)), mechanical properties (tensile strength and elongation at break), and optical parameters (*L**, *a**, and *b** and opacity) of whey–gelatin films with or without incorporation of *L. brevis* SJC120 are shown in Table 1.

The thickness of the whey–gelatin films was significantly (*p* < 0.05) affected by the incorporation of *L. brevis* SJC120 cells, with a lower value ranging from 0.157 mm (control films) to 0.189 mm (film with *L. brevis* SJC120). Ebrahimi et al. [68] also observed an increase in the thickness of carboxymethylcellulose-based films incorporated with probiotic cells compared to the control film. Sogut et al. [23] also reported that the incorporation of various probiotic bacteria significantly affected the thickness of edible films based on whey protein isolate and carrageenan. This higher thickness value can be attributed to the use of a fermented whey, which results in a higher proportion of total solids in the film-forming solution.

The moisture content of edible films is an important parameter, as it plays an important role in the stability of the films. It provides favorable conditions for maintaining the viability of the LAB cells and is important for the easy dissolution of the film when consumed [69]. In the present study, moisture content values showed no difference (*p* > 0.05) between the two films (Table 1). A similar moisture content (25%) was observed in whey protein concentrate incorporated with *L. buchneri* [46]. These results are also in agreement with the observation of Ma et al. [50], who reported that incorporation of *L. lactis* (23.57%) did not significantly affect the moisture content of sodium alginate-based films.

No differences were found in solubility between the films, either (*p* > 0.05) (Table 1). The solubility values were in agreement with previous studies on whey–gelatin films [34]. Other authors also reported no differences in the solubility of films to which LAB strains were added [50,70].

Regarding barrier properties, incorporation of bacterial cells into the film did not significantly change WVTR and WVP (*p* > 0.05) (Table 1). Lower WVP may be a useful property of edible films in a food, as it prevents the loss or increase of moisture transfer [71] and is also necessary to control undesirable chemical reactions and structural deterioration that promote microbial spoilage of food [46]. Similar behavior to WVP was also found for limonene permeability, as no significant differences were found between film formulations (*p* > 0.05). Previous studies have also reported that the addition of LAB cells does not cause major changes in the barrier properties of films containing bacterial cells [26,27,34,46,72].

The mechanical properties of edible films are measured as tensile strength (maximum tensile stress when stretched) and elongation at break (increase in film length to the point of breaking). An edible film must withstand normal stresses during its application to food and maintain its integrity [73]. The results of tensile strength and elongation at break of the films incorporated with *L. brevis* SJC120 are shown in Table 1. For tensile strength, the presence of *L. brevis* SJC120 cells had no significant effect (*p* > 0.05) compared to the control film. These results agree with other studies where the incorporation of bacterial cells had no effect on the tensile properties of the films [26,27,46,63]. The values for tensile strength of the whey–gelatin films (control films) are also similar to those of Silva et al. [34], but higher than those obtained for cheese whey and cassava peel starch films (0.97 N/mm^2^) [74]. However, the elongation at break was significantly higher (*p* < 0.05) for whey–gelatin films containing *L. brevis* SJC120 than for control films, although both films had high extensibility. A similar increase in extensibility was observed by Kanmani et al. [69] for films of pullulan and corn starch films with incorporated LAB cells. 

Optical properties are important in defining the ability of films to be applied to a food surface. They are evaluated by their gloss, color, and transparency [63]. The hue values (*L**, *a**, *b**) and opacity of bioactive films are shown in Table 1. Only a few parameters (b* and opacity) were significantly (*p* < 0.05) affected by the incorporation of *L. brevis* SJC120 cells, and all samples showed a yellowish color typical of cheese whey [34]. After casting, the films were found to be flexible, homogeneous, transparent, glossy, and had a slightly yellowish color. In general, both films have the same luminosity (*L**), the control films showed a slightly greenish hue (*a**) than films with *L. brevis* SJC120 (*p* < 0.05), and both films have a yellowish hue (*b**). The yellowish color of the films was similar to that observed by other authors in films containing fermented cheese whey [34,74]. Furthermore, both films showed no difference (*p* > 0.05) in the transparency of the film.

### 3.5. Antifungal Activity of Films Used in Cheese

Whey–gelatin films with or without *L. brevis* SJC120 were applied to model cheddar-type cheeses to evaluate the effectiveness of inhibiting fungal growth. Cheeses spiked with *Penicillium* species on the surface was evaluated for fungal growth during storage at 10 °C for 30 days (Figure 5).

In the control films, fungal colonies of *P. brevicompactum* and *P. commune* were visible after 5 days of storage, whereas no growth was observed in the *L. brevis* SJC120 films (Figure 5A,B). On day 5, growth of *P. nordicum* was not visible in any of the films (Figure 5C). The cheeses wrapped with the *L. brevis* SJC120 films significantly reduced the growth of filamentous fungi from day 10 to the end of the storage period (30 days) (*p* < 0.05). On the 15th day of ripening, the infection process was considerably delayed in cheese samples wrapped with *L. brevis* SJC120 films compared to cheeses wrapped with control films (Figure 6). At day 15, application of whey gelatin films containing *L. brevis* SJC120 reduced the size of fungal colonies of *P. brevicompactum*, *P. commune*, and *P. nordicum* by 76%, 72%, and 55%, respectively. At the end of the maturation period (30 days), the film containing *L. brevis* SJC120 reduced the size of fungal colonies of *P. brevicompactum* by 73%, while the growth of *P. commune* and *P. nordicum* was reduced by 55%.

The number of studies using antifungal LAB strains in films is quite limited, especially in relation to cheese products. Dopazo et al. [35] reported an effective reduction with a polyvinyl alcohol (PVOH) film containing cell-free fermented whey of *L. plantarum*, with an extension of the shelf life of cheese slices by 15 days for samples contaminated with *P. commune*, by 13 days for samples contaminated with *P. verrucosum*, and by 14 days for samples contaminated with *P. solitum*. Guimarães et al. [46] observed complete inhibition of *P. nordicum* after 30 days of storage by applying films of whey protein concentrate containing *L. buchneri* to cheese, but these authors used a 30% concentration of the strain LAB compared to a 10% concentration of *L. brevis* SJC120 used in the present study. Other studies have reported the use of different compounds in antifungal films or coatings. Among these, natamycin is the most commonly used compound incorporated into the films. In studies such as that of Ture et al. [75], a 2-Log reduction in the spore count of *A. niger* was observed in fresh Kashar cheese coated with methylcellulose containing natamycin, while Fajardo et al. [76] reported that cheese coated with natamycin had a 1.1 Log CFU/g reduction in molds compared to the control after 27 days of storage.

### 3.6. Effect of Film Application on Cheese Ripening

Films (with or without LAB cells) were applied to model cheddar cheeses to evaluate their influence on the properties of the cheese after maturation. The chemical composition of the cheese samples after 30 days of maturation is shown in Table 2.

The values for total fat, protein, and moisture content at the end of maturation did not differ between the cheeses without and with both films. These results are in agreement with those obtained by other authors for the production of cheddar cheese after 30 days of ripening [72]. As expected, the moisture content decreased during maturation for all cheese samples. Similar results were reported by Henriques et al. [77] for hard cheeses coated with antimicrobial edible coatings of whey protein.

The influence of the films on the proteolysis of cheese during ripening was assessed as soluble N in WSE and free amino acids (Table 2). Proteolysis of cheese occurs initially through the direct action of rennet or other indigenous proteolytic enzymes, which produce a series of large- and medium-sized peptides from caseins (primary proteolysis), which are then hydrolyzed by proteinases to shorter peptides and amino acids (secondary proteolysis) [78]. As expected, the soluble N content in the WSE fraction increased (*p* < 0.05) during ripening in all cheese samples, but no differences (*p* > 0.05) were found between treatments. The use of different films also had no effect (*p* > 0.05) on free amino acids, although these also increased with ripening time. Sallami et al. [79] reported that the WSE content and free amino acid content in cheddar cheese increased as ripening progressed when *L. lactis* strains were used as starter culture. Picque et al. [80] also observed a linear increase in nitrogen fractions of cheese wrapped with a film of biodegradable material, polylactic acid, and paraffin of plant origin after 23 days of ripening. In the present study, the similarity of the profiles of the nitrogen fractions obtained for cheese without or with *L. brevis* SJC120 whey–gelatin films and uncovered cheese indicates that the molecular and chemical interactions during cheese proteolysis have the same conditions in all treatments. This suggests that whey–gelatin films containing *L. brevis* SJC120 can be applied to semi-hard cheeses without affecting proteolysis and thus aroma and sensory properties.

The texture characteristics of cheeses were evaluated by the fracturability (breaking force) and are shown in Table 2. The cutting strength increased for all cheese samples during maturation. At the beginning of ripening, the cheese was soft and more elastic than at the end of ripening, where it was harder and more brittle. It was also found that the presence of a film without or with *L. brevis* SJC120 had no effect on hardness. Cerqueira et al. [81] pointed out that cheese with lower moisture content is harder. This agrees with our results, as cheeses with higher moisture content had lower cut hardness values during the first 15 days of ripening (Table 2). Similar effects on cheese hardness were found by Henriques et al. [77] with a whey protein coating on semi-hard cheeses.

The color analysis based on the *L**, *a**, and *b** coordinates is shown in Table 3 and confirms that the color of all cheese samples changed during storage (*p* < 0.05). The strongest change in lightness (*L**) occurred during the first 15 days of storage. A similar decrease in *L** was observed in processed cheese coated with sodium caseinate and 1% inulin in combination with a *Bifidobacterium* strain after being stored for 30 days [82].

At the beginning of ripening, the cheese had an initial light-yellow color, which then changed to a more intense yellow color, which is consistent with the *b** values. For cheeses wrapped in either film, the changes in *a** and *b** values are less pronounced due to the whey–gelatin film. Bonilla et al. [83] also showed similarity of *a** (<7) and *b** values (28–36) between uncovered Prato cheese and cheese wrapped in pure and mixed gelatin and chitosan films, maintaining the typical yellow-orange color during the storage period. However, after removing the whey–gelatin films (control and containing *L. brevis* SJC120), the *a** and *b** values were similar to those of the uncovered cheeses. Color changes in cheese may be due to oxidation caused by oxygen and light. Our results show that whey–gelatin films without or with *L. brevis* SJC120 have no influence on the color differences, even though the opacity values of the films were higher.

During the 30-day ripening, the opacity of the films increased, which could explain the *L**, *a**, and *b** values of the cheeses wrapped with films. This increase in film opacity was observed in our previous study, which can be explained by the fact that peeling off the films after molding may lead to some stretching and arrangement of the protein networks [34].

## 4. Conclusions

In this study, whey–gelatin films were prepared containing cheese whey fermented by five LAB with antifungal properties. Of the LAB tested, *L. brevis* SJC120 was most active at both 10 °C and 20 °C against five filamentous fungi isolated from the cheese-making environment. It was found that the film formulation supported the viability of this strain for at least 30 days. Moreover, the presence of *L. brevis* cells did not affect the main properties of the film, such as moisture, WVP, and mechanical properties (TS and EAB). By applying the whey–gelatin films to a cheddar cheese model, it was confirmed that the films containing *L. brevis* SJC120 greatly reduced the growth of filamentous fungi during the 30-day ripening period. Furthermore, no changes in proteolysis and cheese color were observed by using the film during cheese ripening. Overall, the use of *L. brevis* SJC 120 as a bioprotective culture in whey–gelatin films for cheese packaging showed promising results in solving a problem of the dairy industry, namely finding new approaches to prevent or reduce fungal contamination and reusing cheese whey for a simple and useful alternative to petroleum-derived plastics for cheese packaging.

## Figures and Tables

**Figure 1 foods-12-01396-f001:**
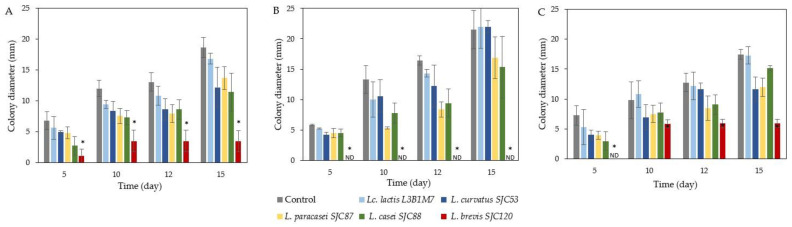
Effect of whey–gelatin films incorporated with *Lc. lactis* L3B1M7, *L. curvatus* SJC53, *L. paracasei* SJC87, *L. casei* SJC88, and *L. brevis* SJC120 on fungal growth (mm) of (**A**) *P. brevicompactum*, (**B**) *P. commune*, and (**C**) *P. nordicum* for 15 days at 10 °C. Values are the mean ± SEM (*n* = 3). * *p* < 0.05 versus control. ND: not detected (no fungal growth observed).

**Figure 2 foods-12-01396-f002:**
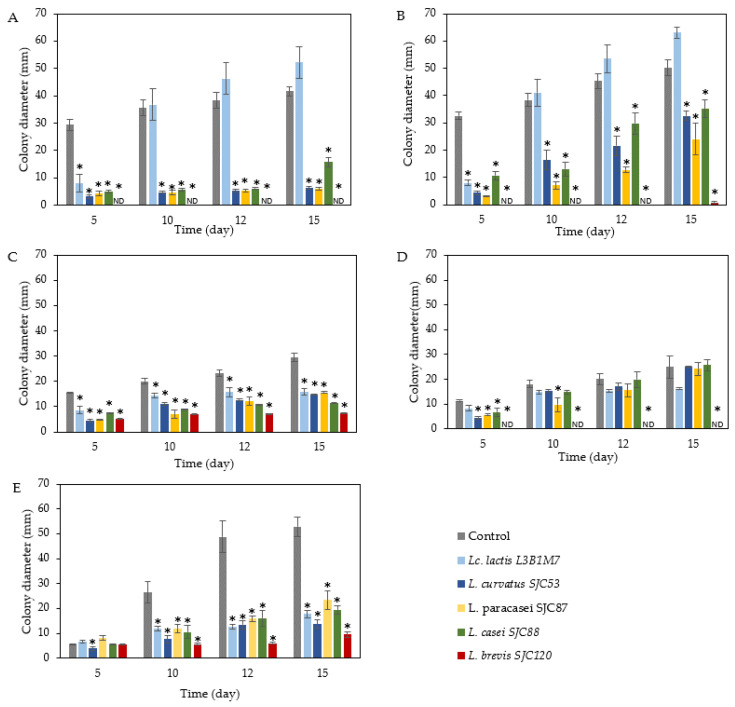
Effect of whey–gelatin films incorporated *L. lactis* L3B1M7, *L. curvatus* SJC53, *L. paracasei* SJC87, *L. casei* SJC88, and *L. brevis* SJC120 on fungal growth (mm) of (**A**) *A. chevalieri*, (**B**) *A. flavus*, (**C**) *P. brevicompactum*, (**D**) *P. commune*, and (**E**) *P. nordicum* for 15 days at 20 °C. Values are the mean ± SEM (*n* = 3). * *p* < 0.05 versus control. ND: not detected (no fungal growth observed).

**Figure 3 foods-12-01396-f003:**
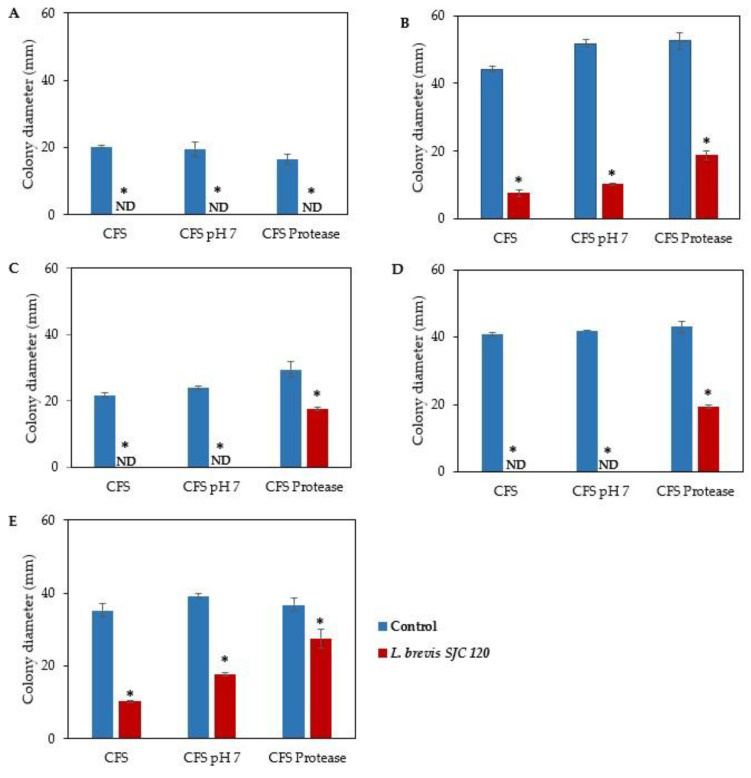
Effect of CFS from L. brevis SJC120 on fungal growth (mm) of (**A**) *A. chevalieri*, (**B**) *A. flavus*, (**C**) *P. brevicompactum*, (**D**) *P. commune*, and (**E**) *P. nordicum*. CFS: untreated CFS, CFS pH 7: neutralization of CFS, and CFS protease: CFS treated with protease K. Values are the mean ± SEM (*n* = 3). * *p* < 0.05 versus control. ND: not detected (no fungal growth observed).

**Figure 4 foods-12-01396-f004:**
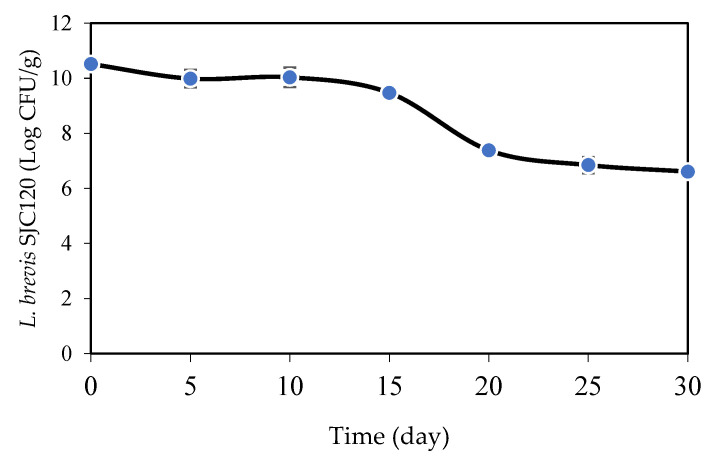
Number of viable *L. brevis* SJC120 (expressed as Log CFU/g) in whey–gelatin films during storage time (day) at 10 °C. Values are the mean ± SEM (*n* = 3).

**Figure 5 foods-12-01396-f005:**
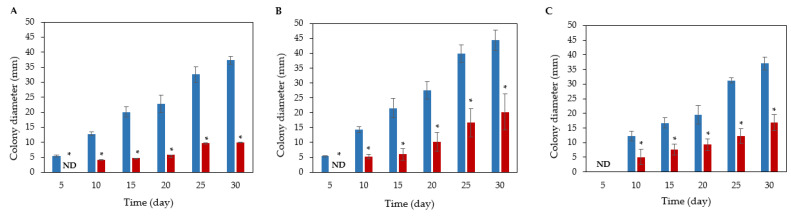
Effect of whey–gelatin films incorporated with L. brevis SJC120 and applied to cheese on fungal growth (mm) of (**A**) *P. brevicompactum*, (**B**) *P. commune*, and (**C**) *P. nordicum* for 30 days at 10 °C. The blue bars represent control films (without LAB), and the red bars represent *L. brevis* SJC120 films. Values are the mean ± SEM (*n* = 3). * *p* < 0.05 versus control. ND: not detected (no fungal growth observed).

**Figure 6 foods-12-01396-f006:**
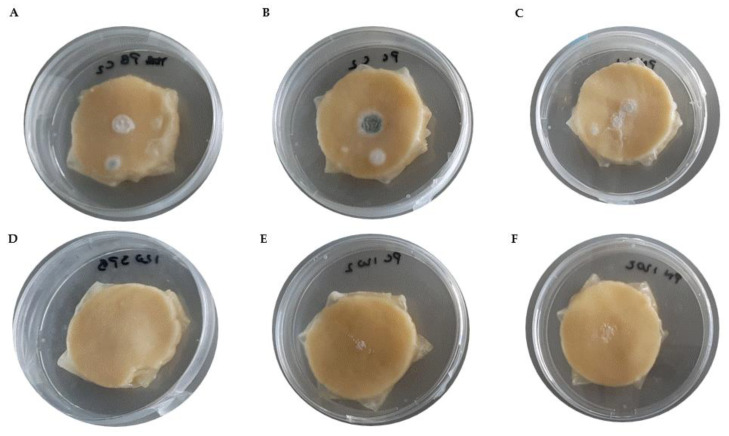
Growth on day 15 of *P. brevicompactum* (**A**,**D**), *P. commune* (**B**,**E**), and *P. nordicum* (**C**,**F**) by inoculation with spore solution on the surface of cheese wrapped in whey–gelatin control films (**A**–**C**) and whey–gelatin films containing *L. brevis* SJC120 (**D**–**F**).

**Table 1 foods-12-01396-t001:** Thickness, moisture content, solubility, water vapor transmission rate (WVTR), water vapor permeability (WVP), limonene permeability (LP), tensile strength, elongation at break, color parameters: *L**, *a**, and *b** and opacity of whey–gelatin films without (control) and with *L. brevis* SJC120. Values are the mean ± SEM (*n* = 3).

	Control Film ^1^	Film with *L. brevis* ^1^
Thickness (mm)	0.157 ± 0.009 ^a^	0.189 ± 0.012 ^b^
Moisture content (%)	27.2 ± 0.5 ^a^	28.3 ± 0.5 ^a^
Solubility (%)	56.6 ± 3.7 ^a^	60.3 ± 2.6 ^a^
WVTR (g/hm^2^)	8.56 ± 1.51 ^a^	8.20 ± 2.05 ^a^
WVP (g mm/m^2^ h kPa)	0.542 ± 0.096 ^a^	0.742 ± 0.173 ^a^
LP (g mm/m^2^ h kPa)	0.242 ± 0.084 ^a^	0.280 ± 0.056 ^a^
Tensile strength (N/mm^2^)	1.836 ± 0.115 ^a^	1.925 ± 0.036 ^a^
Elongation at break (%)	211.7 ± 1.8 ^a^	232.3 ± 1.7 ^b^
*L**	39.970 ± 0.147 ^a^	39.429 ± 0.127 ^a^
*a**	−0.746 ± 0.023 ^a^	−0.527 ± 0.035 ^b^
*b**	1.379 ± 0.091 ^a^	0.979 ± 0.091 ^a^
Opacity (UA/mm)	0.839 ± 0.084 ^a^	0.895 ± 0.008 ^a^

^1^ Different lowercase letters as superscripts in the same line indicate a significant difference (*p* < 0.05) between films.

**Table 2 foods-12-01396-t002:** Evaluation of moisture content, fat, protein, water soluble extract (WSE), free amino acids (FAA), and fracturability (breaking force) of cheddar-type cheese without film (uncovered), covered with control films (without LAB), and covered with *L. brevis* SJC120 films. Values are the mean ± SEM (*n* = 3).

Cheese Samples	Ripening (Day)	Fat (%)	Protein (%)	Moisture (%)	WSE (%)	FAA (mM Leu)	Fracturability (N)
Uncovered *	0			50.3 ± 0.6 ^a^	0.381 ± 0.061 ^a^	1.490 ± 0.255 ^a^	9.96 ± 0.28 ^a^
	15			37.7 ± 1.8 ^b^	0.690 ± 0.019 ^b^	2.700 ± 0.080 ^ab^	17.61 ± 1.79 ^b^
	30	28 ± 1.2 ^a^	26.4 ± 3.7 ^a^	35.4 ± 1.3 ^b^	0.845 ± 0.030 ^c^	2.880 ± 0.232 ^b^	28.58 ± 0.47 ^b^
Control film	0			50.9 ± 1.4 ^a^	0.381 ± 0.061 ^a^	1.471 ± 0.141 ^a^	9.42 ± 0.68 ^a^
	15			37.1 ± 2.1 ^b^	0.576 ± 0.019 ^b^	2.116 ± 0.105a ^b^	17.26 ± 1.75 ^b^
	30	25 ±1.5 ^a^	26.8 ± 1.3 ^a^	35.7 ± 1.2 ^b^	0.954 ± 0.079 ^c^	2.723 ± 0.241 ^b^	28.04 ± 1.20 ^b^
*L. brevis* film	0			50.3 ± 1.4 ^a^	0.381 ± 0.061 ^a^	1.538 ± 0.018 ^a^	9.27 ± 0.42 ^a^
	15			37.7 ± 2.4 ^b^	0.577 ± 0.028 ^b^	2.199 ± 0.098 ^ab^	17.61 ± 1.79 ^b^
	30	27 ± 1.1 ^a^	27.3 ± 1.3 ^a^	35.7 ± 0.4 ^b^	0.806 ± 0.030 ^c^	2.318 ± 0.168 ^b^	28.66 ± 0.62 ^b^

* Different lowercase letters as superscripts in the same column indicate significant differences (*p* < 0.05).

**Table 3 foods-12-01396-t003:** Evaluation of film opacity and optical properties (*L**, *a** and *b**) of cheeses during maturation at 10 °C for 30 days. Optical properties of cheeses are presented without film (uncovered), wrapped with control film, wrapped with *L. brevis* SJC120 films, and after removing of films. Values are the mean ± SEM (*n* = 3).

Cheese Samples	Ripening (Day)	*L**	*a**	*b**	Opacity * (UA/mm)
Uncovered	0	86.276 ± 0.274	3.509 ± 0.113	34.022 ± 0.761	na
	15	69.444 ± 0.437	8.018 ± 0.076	38.295 ± 0.531	na
	30	69.655 ± 0.775	7.143 ± 0.131	43.093 ± 0.270	na
Wrapped with film					
Control film	0	83.356 ± 1.339	4.301 ± 0.180	39.477 ± 0.505	0.865 ± 0.049
	15	74.622 ± 0.896	4.367 ± 0.390	25.629 ± 1.560	3.893 ± 0.148
	30	76.786 ± 0.738	2.966 ± 0.296	20.155 ± 0.991	6.571 ± 0.384
*L. brevis* film	0	83.715 ± 0.583	4.069 ± 0.268	33.383 ± 1.114	0.844 ± 0.078
	15	73.125 ± 0.639	4.168 ± 0.286	29.770 ± 1.422	3.664 ± 0.505
	30	78.332 ± 1.563	2.639 ± 0.232	17.941 ± 1.802	6.687 ± 0.499
After film removed					
Control film	15	69.790 ± 2.183	6.898 ± 0.040	37.776 ± 0.529	na
	30	67.320 ± 0.354	6.582 ± 0.082	43.660 ± 0.226	na
*L. brevis* film	15	70.040 ± 0.411	7.656 ± 0.072	40.792 ± 0.072	na
	30	67.530 ± 0.857	7.298 ± 0.046	42.126 ± 0.291	na
Two-way ANOVA	Cheese samples	*p* < 0.001	*p* < 0.001	*p* < 0.001	*p* > 0.05
	Time	*p* < 0.001	*p* < 0.001	*p* < 0.001	*p* < 0.001
	Interaction	*p* < 0.001	*p* < 0.001	*p* < 0.001	*p* > 0.05

* na: not applicable.

## Data Availability

Data available from the corresponding author.

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
