# Peer review of "Prevention of Fungal Contamination in Semi-Hard Cheeses by Whey–Gelatin Film Incorporated with Levilactobacillus brevis SJC120"

_foods, 2023, doi:10.3390/foods12071396_

Round 1

Reviewer 1 Report

I have read carefully the paper entitled “ Prevention of fungal contamination in semi-hard cheeses by whey-gelatin film incorporated with Levilactobacillus brevis SJC120”. This study is carefully conducted in respect of the increasing need for the antimicrobial strategy of dairy products. A very strong aspect of this study is the variety of aspects in antimicrobial analyses that were reviewed. However, there are some issues with the text and the interpretation of the results that need revision. Overall, my suggestion is that the paper should with minor revisions.

English language must be corrected throughout the whole text. It is inappropriate to use phrases and active forms of sentences. Please, provide better phrase works for each paragraph in the paper.

The second very important thing: the visualization of all graphs and figures have to be changed, not only in quality but also in better interpretation of the collected results. Everything is very minimised and looking at them is tiring. All figures need more creativity and information to be better for further use

The study lacks an understanding interaction of fungal activity/growth/contamination and decreasing quality of cheese. I need to require more information about the microbiology aspect in this study since the presence of microorganisms can decrease the quality of fresh-cut produce and reduction of the expiration date.

Author Response

English language must be corrected throughout the whole text. It is inappropriate to use phrases and active forms of sentences. Please, provide better phrase works for each paragraph in the paper.

  • We thank the referee for the commentary and have thoroughly revised the manuscript to change phrases from active to passive. The revisions in the manuscript are highlighted in red and were made with the help of an English language expert.

The second very important thing: the visualization of all graphs and figures have to be changed, not only in quality but also in better interpretation of the collected results. Everything is very minimised and looking at them is tiring. All figures need more creativity and information to be better for further use

  • We thank the reviewer for the comments and understand that they contain a lot of information. For clarity, we have increased the scale of the axes to make the graphs easier to read. We had also presented a separate graph for each fungal species to make the figures more understandable. However, space in the manuscript is limited, so we provided high-quality images separately.

The study lacks an understanding interaction of fungal activity/growth/contamination and decreasing quality of cheese. I need to require more information about the microbiology aspect in this study since the presence of microorganisms can decrease the quality of fresh-cut produce and reduction of the expiration date.

  • We thank the reviewer for the suggestions and have rewritten the introduction section to include more information about fungal spoilage of cheese and loss of product quality – Lines 28-40.

Reviewer 2 Report

There are too many paragraphs in the introduction section, please briefly summarize the progress in the field and cite recent related literature.

The first paragraph of Lines 28-35 is unnecessary.

Line 37. Also including natamycin. Ref also including natamycin such as [International Journal of Biological Macromolecules 209, 2042-2049].

Line 42-57. Please simplify and combine paragraphs with similar meaning expressions.

Line 58-62. Please merge with the next paragraph. Ref 26, 27 are also recommended adding recent Journal papers with antifungal films and coatings, such as [Food Packaging and Shelf Life 2023, 35,101019; Food Hydrocolloids 2021, 113, 106506.]

Line 73. Why gelatin films should be prepared instead of other polymers such as chitosan. Please explain briefly.

line 85-90. Is there any evidence that these strains can inhibit fungi? Please provide literature.

Section 3.1 What are the antibacterial substances? Is there any literature describing the possible compounds and the minimum inhibitory concentration data?

Line 339-346. Please keep in mind that some secretions with anti-fungal effects are cyclic peptide antibiotics and have strong side effects.

Figure 4 is suggested to be redone for a better look.

Author Response

There are too many paragraphs in the introduction section, please briefly summarize the progress in the field and cite recent related literature.

  • R. We thank the reviewer for his comment and have revised the introduction to better summarize the progress in this area.

The first paragraph of Lines 28-35 is unnecessary.

  • R. We have retained this paragraph, albeit with some changes to address Referee 1's comments.

Line 37. Also including natamycin. Ref also including natamycin such as [International Journal of Biological Macromolecules 209, 2042-2049].

  • R. We thank the reviewer for the suggestion and have included natamycin and the reference.

Line 42-57. Please simplify and combine paragraphs with similar meaning expressions.

  • R. We thank the reviewer for the suggestions and have rewritten the introduction section.

Line 58-62. Please merge with the next paragraph. Ref 26, 27 are also recommended adding recent Journal papers with antifungal films and coatings, such as [Food Packaging and Shelf Life 2023, 35,101019; Food Hydrocolloids 2021, 113, 106506.]

  • R. We thank the reviewer for the suggestions and have included the references in line 58.

Line 73. Why gelatin films should be prepared instead of other polymers such as chitosan. Please explain briefly.

  • R. We thank the reviewer for the suggestion and have included this statement in the introduction (lines 56-60): “Many types of polymeric materials have been used in the development of edible films and coatings with various functional properties, especially as effective carriers for bioactive compounds [31-33]. Among the various formulations, collagen (gelatin) and whey proteins have been widely used for incorporating live bacteria such as LAB into edible packaging [32,34-36].”

line 85-90. Is there any evidence that these strains can inhibit fungi? Please provide literature.

  • R. We have a manuscript in preparation describing the antifungal activity of several LAB with antifungal activity, including these strains. Since the manuscript is not yet published, we have included the reference to a PhD thesis (line 91).

Section 3.1 What are the antibacterial substances? Is there any literature describing the possible compounds and the minimum inhibitory concentration data?

  • R. We believe the reviewer means the antifungal compounds. We do not yet know which compounds are responsible for the antifungal activity. This will be investigated in further studies.

Line 339-346. Please keep in mind that some secretions with anti-fungal effects are cyclic peptide antibiotics and have strong side effects.

  • R. We use lactobacilli species, especially Levilactobacillus brevis, which, as far as we know, do not produce these cyclic peptide antibiotics that can be toxic. For this reason, these bacteria have the QPS status granted by the European Food Safety Authority (EFSA).

Figure 4 is suggested to be redone for a better look.

  • R. We thank the reviewer for the suggestion and have revised Figure 4.